# Quality Characteristics Analysis and Remaining Shelf Life Prediction of Fresh Tibetan *Tricholoma matsutake* under Modified Atmosphere Packaging in Cold Chain

**DOI:** 10.3390/foods8040136

**Published:** 2019-04-22

**Authors:** Zetian Fu, Shuang Zhao, Xiaoshuan Zhang, Martin Polovka, Xiang Wang

**Affiliations:** 1Beijing Laboratory of Food Quality and Safety, College of engineering, China Agricultural University, Beijing 100083, China; fzt@cau.edu.cn (Z.F.); zhaosh@cau.edu.cn (S.Z.); zhxshuan@cau.edu.cn (X.Z.); 2National Agricultural and Food Centre Priemyselná, Radlinského 9, SK-81237, Slovak Republic; polovka@vup.sk

**Keywords:** Tibetan *T. matsutake*, quality characteristics, shelf life, modified atmosphere packaging, cold chain

## Abstract

*Tricholoma matsutake (T. matsutake)* growing in Tibet is very popular for its high economic and medicinal value, but fresh *T. matsutake* has an extremely short shelf life. The shelf life of *T. matsutake* is complex, influenced by product characteristics, surrounding environmental conditions, and spoilage development. The objective of this work was to study the quality characteristics of fresh *T. matsutake* during its shelf life period in modified atmosphere packaging (MAP) conditions and establish its remaining shelf life prediction models in a cold chain. In this study, we measured and analyzed quality indicators of fresh *T. matsutake*, including hardness (cap, stipe), color, odor of sensory characteristics, pH, soluble solids content (SSC), and moisture content (MC) of physical and chemical characteristics under the temperature condition of 4 °C and relative humidity (RH) of 90%. The sensory evaluation results showed that the odor indicator in sensory characteristics was more sensitive to the freshness of *T. matsutake*. The changes of pH, SSC, and MC were divided into three periods to analyze the physiological changes of *T. matsutake*. The cap spread process could affect the changes of pH, SSC, and MC in period S_1_, and they changed gradually in period S_2_. In the period S_3_, they changed complicatedly because of deterioration. The remaining shelf life prediction model of *T. matsutake* was established by the back propagation (BP) neural network method to quantify the relationship between the quality indicators and the remaining shelf life. The shelf life characteristics are complex, which were optimized by correlation analysis. Significant benefits of this work are anticipated on the transportation and preservation of fresh *T. matsutake* to the market and the reduction of its losses in the postharvest chain.

## 1. Introduction

*T. matsutake* grown in different areas has different qualities. In China, *T. matsutake* in Yunnan Province is the most famous, and the quality of *T. matsutake* in Tibet is the best [1]. Tibet has rich species, a variety of vegetation types, and superior animal and fungi sources because of its serene nature. The average altitude of the Tibet Autonomous Region is over 4000 m. It is located in the borderland, and has transport problems, leading to long logistics paths and high cost. The development of edible fungi industry in Tibet is faced with the challenge of extending the shelf life and minimizing post-harvest food and economic losses [2]. *T. matsutake* has a high reputation in Japan and South Korea [3], and it has always been admitted as a treasure in the edible mushroom family, and has been called the king of mushrooms [4]. It was a kind of precious wild edible fungi with its strong fragrance, stout body, delicate flesh, and special flavor, and took the fancy of consumers for its high nutritional and medicinal value [5]. As a kind of wild edible fungus, *T. matsutake* has a relatively short life when compared with other edible funguses, such as *mushroom,* white *Hgpsizygus marmoreus* and *Pleurotus ostyeatus* [6,7,8], due to high respiration rate and transpiration, which significantly decrease its marketing period [9,10]. There is no protective structure on the surface of *T. matsutake*, which makes it highly susceptible to infection and then decay [9]. The shelf life of *T.matsutake* is about 2 days after harvest, when it is preserved under the temperature of about 20 °C [11]. Then it happens cap opening, browning, autolysis, and other phenomena, and loses its commodity and edible value [3]. Its high fragility and its very short shelf life is the main obstacle to its transportation and consumption, leading to considerable waste and loss during the post-harvest steps [2].

The cold chain can postpone the quality loss of edible fungi to increase its shelf life, but it is far from being able to meet the industry needs [12]. Modified atmosphere packaging (MAP) technology has been widely used for the shelf life extension of edible funguses [13,14,15], fruit, and vegetables [16], which is a promising technique to delay the deterioration of *T. matsutake* in the post-harvest chain. MAP can achieve an optimal gas composition in the close environment of the product [16,17,18]. The modified atmosphere condition slows down the respiration rate, ethylene production, and water loss, thus reducing the enzyme activities and metabolic rate of fresh product [19,20,21,22]. In recent years, MAP has proved an effective method to modify the physiology and prolong the shelf-life of fresh food by flushing the desired initial gas into the packages in a cold chain [23,24]. *T. matsutake* is living organism; it continues to respire after harvest by absorbing oxygen from air and producing carbon dioxide. After a transition phase, a gas equilibrium is established around the product, of which the composition must be as close as possible to the optimal one to reduce respiration, prevent ripening, senescence, and fermentation, and thus increase shelf life [16,25].

Although significant benefit in terms of food loss reduction is expected from shelf life extension, this direct positive effect is difficult to anticipate for the post-harvest step [26]. This is due to the lack of generalized quality indicators to quantify the shelf life of edible fungi products in general, especially in *T. matsutake*. The shelf life of *T. matsutake* is complex, influenced by product characteristics (weight loss rate, browning index, pH, soluble solids content, etc.), surrounding environmental conditions (temperature, gas composition, and relative humidity of the atmosphere), and spoilage development [4,9,27]. Many quality parameters, such as firmness, chromatic aberration, pH, respiratory intensity, soluble solids content (SSC), browning index, weight loss rate, and sensory evaluation have been considered to be the main shelf life influence factors of *T. matsutake* [3,11]. It is not certain which indicators directly affect its quality and shelf life. Therefore, it is essential to analyze the shelf life parameters of *T. matsutake* and select the main factors to quantify its shelf life quality. Correlation analysis is a statistical method used to evaluate the strength of relationships between different variables [28]. Zhang et al. found that the hardness, protein, total sugar, chromatic aberration (L*), and pH of *Pleurotus eryngii* mushroom displayed a significantly positive correlation with sensory quality score using the method of correlation analysis [14]. Fu et al. used correlation analysis to study the relationship between gas and physical and chemical indexes on shelf life of blueberry [29]. In this paper, we adopt correlation analysis to optimize the quality indicators on shelf life of *T. matsutake* by analyzing the relationship between the quality indexes data with its remaining shelf life.

Research has been done on predicting the shelf life of edible fungi. Researchers discussed the shelf life prediction models of edible mushrooms, such as *Shiitake* mushroom, by using a kinetic model [26,30] based on temperature parameters, or statistics models [6,14]. However, there are few studies predicting the shelf life of edible fungi based on the quality indicators under the MAP condition. With the development of science and technology, the back propagation (BP) neural network has become the most widely used prediction model, which is a multi-layer feedforward one-way propagation network based on error feedback and using a backward propagation algorithm [31]. Compared with the traditional statistical methods, the BP neural network can correct the equation according to the quality index of the food and obtain the mathematical equation of the appropriate product to predict the shelf life [32]. The BP neural network has been applied to the shelf life prediction of fruit and vegetables, such as blueberry [29], fresh egg [33], and some food like quick-frozen dumpling [34].

Based on the above discussion, this paper aims to study the quality indicators of fresh Tibetan *T. matsutake* changes and build its remaining shelf life prediction model through optimizing the quality characteristics of *T. matsutake* in different MAP under the temperature condition of 4 °C and relative humidity of 90%. The data in the experiment of quality indicators were analyzed to determine the variation tendency. The remaining shelf life prediction model of *T. matsutake* was established by the BP neural network method. Correlation analysis was used to optimize the indicators affecting the shelf life by the relativity between quality indicators and the remaining shelf life of *T. matsutake*. It could provide a reference and help for the preservation, transportation and sales of *T. matsutake*.

## 2. Materials and Methods

### 2.1. Raw Material

Fresh *T. matsutake* were obtained from the forest area of Lingzhi City, Tibet Autonomous Region. *T. matsutake* were harvested in the morning of the experiments, cooled to about 5 °C in the incubator, then transported to the laboratory under refrigerated conditions (4 ± 1 °C) and 90% relative humidity (RH) around 2 h after harvest. Harvested *T. matsutake* were sorted according to the following standards: Maturity was 80–90%. The cap was smooth and unopened, and the color was shining chestnut-colored. The stipe was not shedding, and its color was ivory. Sorting was done also according to shape. Oversized *T. matsutake* or very small ones compared to the batch *T. matsutake* sizes were eliminated. Finally, damaged or rotten *T. matsutake* were removed [35]. They were cleaned by a soft brush, which was used to remove debris from the *T. matsutake* surface and soil from the root.

After pre-cooling, sorting, and cleaning treatments, the *T. matsutake* were packaged in vacuum fresh keeping bags made of polyethylene (PE) (Weide New-material Corp., Xuzhou, Jiangsu, China) with dimensions of 0.15 × 0.20 m. Each bag contained 2 pieces of *T. matsutake* of about 0.2 kg. The total amount of fresh *T. matsutake* was 10 kg for this experiment.

### 2.2. Packaging, Storage, and Atmosphere Composition

Four packaging conditions were considered in these experiments, and the components are exemplified in Table 1. The temperature of these tests was set at 4 °C and the relative humidity was 90%. The air group was used as a controlled trial, and the other three groups were set with different ratios of O_2_ and CO_2_ in the atmosphere space to do the experiments.

Before closing the fresh-keeping bags, which contained the *T. matsutake* filled with the mix standard gases, we needed to draw the air out of the bags and seal up these bags by a vacuum packaging machine (Deli, Ningbo, Zhejiang, China). A small hole was cut in the corner of each bag, which could only accommodate a gas supply pipe about 8 mm wide. Mixed standard gases bought from a standard gas factory in local (Linzhi, Tibet), were used to fill the fresh-keeping bags. *T. matsutake* with different modified atmosphere packaging are shown in Figure 1.

### 2.3. T. Matsutake Quality Indicators Measurement

In this paper, the data of sensory indicators and physical and chemical indicators of four groups of *T. matsutake* were obtained. Sensory indicators included the hardness (cap, stipe), color, and odor. Physical and chemical indicators included pH, soluble solids content, and moisture content. We measured the quality indicators of *T. matsutake* once every two days. Each time we took 2 pieces and tested them separately, and each indicator was measured three times for the mean value.

Sensory evaluation included the hardness (cap, stipe), color, and odor of fresh Tibetan *T. matsutake*. A trained assessment team panel, including the local gatherer, acquirer, cook, and 2 consumers all of these 5 people who were invited to make the sensory evaluation, observed and filled a form to record the scores separately. In this study, the score of sensory evaluation was divided into four grades; the certain specific standards required to evaluate them are listed in Table 2. In the data processing of sensory score, the final sensory score was the median of evaluation scores by the evaluation team. For example, the odor sensory scores of *T. matsutake* were 3, 3, 2, 3, 2, and then the final odor sensory score was 3.

The pH and soluble solids content (SSC) test procedures were as follows: Put 5 g *T. matsutake* into a mortar that was surrounded with ice. Add 20 mL of phosphate buffer (PB) containing 1% polyvinyl pyrrolidone (PVP) with a pH of 6.8 and a concentration of 0.05 mol/L. The mixture was ground with a pestle in low temperatures, and then filtered the mixture with 4 layers of gauze. The pH value was accessed using a pHB-4 pH meter (INESA.CC, Shanghai, China) by measuring the filtered clear liquid. The content of SSC was gotten by measuring the filtered clear liquid using a glycometer refractometer (LH-B55, Hangzhou, Zhejiang, China).

Moisture content was measured using a moisture meter (Shenzhen crown and moisture meter technology co., LTD., Shenzhen, China) [13]. The weight of each test piece model was 0.05 kg to 0.10 kg.

Correlation analysis was carried out between the data of factors and the remaining shelf life of *T. matsutake*. Experimental data and remaining shelf life data were normalized first, and then Microsoft Excel mathematics software was used for correlation analysis. We selected and recorded the correlation data between the remaining shelf life and various experimental indicators. The greater the correlation coefficient, the more significant relationship between these indexes and their remaining shelf life. The representative indexes were selected as the independent variables and the remaining shelf life was the dependent variable of the shelf life prediction models.

### 2.4. The Remaining Shelf Life Prediction Method—BP Neural Network

BP neural network is a kind of multilayer feed-forward neural network [36], whose main characteristics are the forward propagation of signals and back-propagation of errors. In forward propagation, the input signal is processed layer by layer from the input layer to the hidden layer and then to the output layer. Neurons of each layer only affect the state of neurons of the next layer. If the desired output is not obtained in the output layer, the signal shifts to back-propagation, and the network weight and threshold are adjusted according to prediction error, so that the predicted output of the BP neural network gradually approaches its desired output [37]. The topology of the BP neural network is shown in Figure 2.

As shown in Figure 2, a BP neural network consists of many neurons connected to each other and has the structure of input, hidden, and output layers. In this paper, the input layer parameters were the quality indicators of *T. matsutake*, and the output layer was its remaining shelf life. $w_{\mathrm{ij}}$ and $w_{\mathrm{jk}}$ were weights of the BP neural network. The BP neural network is a multi-layer and feed-forward neural network based on an erroneous reverse transmission algorithm [31,38]. The learning rule of the BP neural network is to use a steepest descent algorithm to continuously adjust the weights and thresholds by the back-propagation network to obtain the minimum sum of squared errors of the network [34]. The BP neural network has to train the network before prediction. The network has association and prediction ability through training. The process of training the network includes network initialization, output calculation of the hidden layer, output calculation of the output layer, error calculation, weight updating, threshold updating, and judgment of whether algorithm iteration has come to an end. It is necessary to go back to step 2 if the algorithm iteration has not come to an end [39,40]. Based on the BP neural network, signals are predicted in the following three steps: construction of the BP neural network, training of the BP neural network, and prediction of the BP neural network.

In general, the input and output layer parameters of the BP neural network have different specifications. In order to reduce the errors of the remaining shelf life prediction model, the input and output layer parameters were normalized according to Formula (1):(1)$$p^{\prime }=\frac{p-p_{min}}{p_{max}-p_{min}}.$$

In Formula (1), $p^{\prime }$ is normalized data, *p* is original data, $p_{min}$ is the minimum of original data, and $p_{max}$ is the maximum of original data.

The number of neurons in the input and output layers of the BP neural network was determined by the input and output layer variables. The calculation of the optimal number of hidden layer nodes was affirmed as follows:(2)l<(m+n)12+a.

In Formula (2), *l* is the number of hidden layer nodes, *n* is the number of input layer nodes, *m* is the number of output layer nodes, and a is a constant between 0 and 10.

In this study, the training parameters of the BP neural network were set as follows: The input layer function was logsig function. The output layer function was purein function. The training function was trainrp function. The learning function was learndm function. The learning rate was set as 0.001. The momentum factor was 0.01. The training error was 1 h. The maximum step size was 10,000 [29]. Matlab R2016b mathematics software (version 9.1, MathWork Inc., Natick, MA, USA) was used to establish the models and analyze the date.

## 3. Results and Discussion

### 3.1. Quality Characteristics Analysis of Fresh Tibetan T. matsutake in MAP Conditions

*T. matsutake* is a kind of respiratory climacteric wild edible fungus. It still continues to live even though there is no water and nutrient supply after harvest. Respiration becomes the dominant process of its metabolism [11]. Low oxygen packaging can reduce the respiration rate and maintain shelf-life longer or with better quality than normal air packaging [22]. Thus, the proportion of oxygen and carbon dioxide concentration in MAP conditions affect its quality and shelf life. In this paper, we set controlled trials: group four (air) was compared with group one (1% O_2_, 21% CO_2_, and 78% N_2_), group two (5% O_2_, 17% CO_2_, and 78% N_2_), and group three (10% O_2_, 12% CO_2_, and 78% N_2_) from 0 to 22 days of storage at 4 °C and 90% RH in polyethylene pouches, to study its shelf life characteristics in different MAP conditions. Sensory characteristics and physical and chemical characteristics of fresh Tibetan *T. matsutake* changes are shown in Figure 3 and Figure 4.

#### 3.1.1. Changes in Sensory Characteristics

In some studies, sensory evaluation was also defined as the consumer willingness to purchase [16]. The limit of acceptability by the consumers, in days, was identified by the time at which less than 50% of purchases occurred, as generally recommended by researchers [41,42]. In this experiment, the shelf life was terminated when more than 50% of the panel considered that the fresh *T. matsutake* did not need to make the sensory evaluation any more, as they had no market value. The shelf life in group one, two, three, and four was 18 days, 14 days, 12 days, and 10 days, respectively. In preset MAP conditions, the shelf life of *T. matsutake* was reduced with the increasing of oxygen concentration in the Figure 3. Furthermore, the sensory evaluation score of these four kinds of indicators decreased with the extension of storage time.

Sensory indicators included the hardness (cap, stipe), color, and odor of fresh Tibetan *T. matsutake*, whose sensory evaluation scores are shown in Figure 3. The odor evaluation decreased more obviously with the increase of time and oxygen, as seen in Figure 3d. Fresh *T. matsutake* has a strong unique fragrance as it grows. Previous studies in the fragrance and freshness of *T. matsutake* suggested that about 2/3 fragrance and 1/3 to 2/3 freshness were lost when they were stored in the air at room temperature for 8 h after collection. After 24 h, only 20% fragrance and freshness remained [5]. Therefore, the odor is the more sensitive indicator for the freshness of *T. matsutake*.

#### 3.1.2. Changes in Physical and Chemical Characteristics

Fresh *T. matsutake* continued to live after harvest. Its life activities were the physiological changes of maturation and senescence on the basis of basic metabolism such as respiration and transpiration. In order to research the physiological changes of fresh *T. matsutake* after harvest, we measured its physical and chemical characteristics, including the pH, soluble solids content (SSC), and moisture content (MC) in this experiment. The biological tissue of fresh *T. matsutake* is tender. After harvest, the moisture content of *T. matsutake* is up to 90%, for protecting its young and tender tissue. However, because of its active metabolism and transpiration, it loses water and consumes nutrients quickly [3].

The changes as a function of time of test indicators are illustrated in Figure 4. According to previous research and data changes, Figure 4 can be divided into three periods, S_1_, S_2_, and S_3_, respectively, to interpret the physiological changes of *T. matsutake*.

As seen in Figure 4a, the changes in pH dramatically increased in the beginning during period S_1_. That is because the cap of *T. matsutake* spread after harvest, like many mushrooms. In this period, the quality declined a lot as a consequence of respiration. Furthermore, compared with traditional preservation methods, MAP delayed its quality change, which sustained more than two days. Then, the changes of pH slowly decreased in period S_2_, which sustained more than eight days. In this period, *T. matsutake* were breathing smoothly. However, the fluctuation of pH suggests a great deal more volatility in period S_3_, which also appeared in period S_3_ of Figure 4b,c, the changes of SSC and MC. With the preservation time flowing, individual quality differences of *T. matsutake* greatly increased. Therefore, the experiment data showed great fluctuation and instability.

In Figure 4b,c, the changes of SSC and MC decreased in the period S_1_, lasting about four days. In period S_2_, the content of SSC remained stable over time, and the MC slowly decreased. Furthermore, in period S_3_, the fluctuation of SSC and MC changed too dramatically to analyze the variation trend. Comparing the changes of SSC and MC in period S_3_, we found that when the line of MC decreased, the line of SSC increased between the tenth day and the fourteenth day. Furthermore, when the line of MC increased, the line of SSC decreased between the fourteenth day and the eighteenth day. We can conclude that there is a correlation between SC and MC of fresh *T. matsutake* in the process of preservation.

In Figure 4c, the MC decreased in the period S_1_ and period S_2_. In the period S_3_, the MC changed differently in every group, whose trends were to increase first and then decrease. In these experiments, individual differences did exist between *T. matsutake*. Except for this reason, *T. matsutake* would experience the autolysis phenomenon after its shelf life, which could make the MC increase. Like in group four, the shelf life had stopped on the tenth day through the sensory evaluation. In days 14–18, we considered that groups two, three, and four had stopped their shelf life, and the pH, SSC, and MC changes in these days were considered as its deterioration periods.

### 3.2. The Remaining Shelf Life Prediction Modelling by BP Neural Network

#### 3.2.1. Parameter Optimization of Prediction Model

The shelf life of *T. matsutake* is complex. In this experiment, sensory indicators, including the hardness (cap, stipe), color, and odor, and physical and chemical indicators, including pH, soluble solids content, and moisture content of *T. matsutake* were tested. It is not very sure which indicators mainly affected its quality and shelf life. Therefore, it is practicable to optimize these indicators to find out the key factors, which can be used in shelf life prediction models as variables. Correlation analysis was used to ascertain the relationship between the indicators measured in the experiment and the remaining shelf life of *T. matsutake*.

Table 3 shows the correlation coefficients between the remaining shelf life and the sensory indexes and physical and chemical indicators of *T. matsutake* in experiments.

According to the analysis results, only three factors among the seven factors tested in the experiment had strong linear correlation with the remaining shelf life. It can be seen in Table 3 that the sensory indicators were positively correlated with the remaining shelf life. Among them, stipe hardness, color, and odor of *T. matsutake* had strong correlations with the remaining shelf life. The other indicators had slight linear relationships under the conditions of this experiment. Therefore, stipe hardness, color, and odor of *T. matsutake* were selected as the independent variables in the prediction model, and the remaining shelf life of *T. matsutake* was the dependent variable in the following prediction models.

#### 3.2.2. BP Neural Network Construction

According to Table 3, the three factors of color, odor, and stipe hardness were selected as the variables predicting the remaining shelf life of *T. matsutake*. Therefore, the input layer of the prediction model had three neurons, which was the number of influencing factors, and the output layer only had one neuron, which was the number of target objects.

The number of neurons in the hidden layer was determined to be 3–12 according to Formula (2). Within this range, a performance test was taken to determine the exact number. The results showed when the number of hidden layer nodes was 10, the network convergence speed was the fastest. Therefore, the number of hidden layer nodes was determined to be 10 in this model.

In this paper, the BP neural network model had a three-layer neural network structure with a single hidden layer. The input layer parameters were stipe hardness, color, and odor; the number of hidden layer nodes was 10; and the output layer parameter was the remaining shelf life of *T. matsutake*. The structure of the BP neural network was 3-10-1.

#### 3.2.3. Prediction Results of the Remaining Shelf Life Prediction Model

Before we used the BP neural network to predict the shelf life of *T. matsutake*, the BP neural network had to be trained first. The data measured in each group was reserved; one group was for predictions, and the rest of the experimental data were substituted into the BP neural network for training. The remaining shelf life of *T. matsutake* was predicted by using the above-mentioned trained BP neural network model. The results of the remaining shelf life under the four experimental conditions are shown in Table 4.

In Table 4, the same results were predicted for several consecutive days: 4th day to 10th in group 1; 4th day to 6th in group 2,3,4, because of the effects of sensory evaluation according to Table 2, *T. matsutake* sensory quality scoring standard. There were only four levels of sensory evaluation of fresh *T. matsutake*, which led to the same value of three sensory indicators in the middle days of the shelf life. Compared with the fluctuation of the physical and chemical index image in Section 3.1, the prediction results showed similar changes: they rapidly decreased in the first four days, and slowly decreased between the fourth day and tenth day. After the tenth day, the prediction results changed a lot.

#### 3.2.4. Prediction Models Verification of BP Neural Network

We took the absolute error to verify the prediction results. As seen in Table 4, the predicted absolute error was over the error range in the fourth day. That was because the data of sensory indicators got their score from the first level to the second level according to the standard of Table 2 in the fourth day. Furthermore, this situation also appeared in Figure 5. From the Figure 5, there was always a red color with the third line in the fourth day. Combined with the analysis of prediction data, the results showed that the quality of *T. matsutake* changed greatly on the fourth day of storage. In addition, there was the possibility of dramatic changes on postharvest characteristics and quality of *T. matsutake* from the second day to the fourth day because of its cap spread process, shown in our analysis on the changes of physical and chemical characteristics in Figure 4. Further specific research is still needed to determine the quality changes of *T. matsutake*. Moreover, the quality changes of *T. matsutake* from the second day to the fourth day should be measured carefully in future research.

## 4. Conclusions

This paper aims to study the shelf life characteristics of fresh Tibetan *T. matsutake* in the MAP conditions under the temperature condition of 4 °C and relative humidity of 90% for improving the shelf life of fresh Tibetan *T. matsutake* and ensuring its quality and safety in shelf life. Correlation analysis was applied to optimize the quality characteristics obtained from the experiments for building the remaining shelf life predicting models of *T. matsutake*. A BP neural network was used to build these models in different MAP conditions to quantify the relationship between quality changes and preservation conditions during shelf life, which provided the research basis for transportation and preservation methods of fresh Tibetan *T. matsutake* in a cold chain.

The experimental results showed that the shelf life of fresh *T. matsutake* reduced with increasing oxygen concentration, when the O_2_ content was between 1% and 21% (the O_2_ content of air), and its shelf life could extend to 18 days under the MAP condition of 1% O_2_ + 21% CO_2_ + 78% N_2_. Sensory characteristics and physical and chemical characteristics were analyzed. Sensory evaluation scores of four kinds of indicators decreased with the extension of storage time. The odor indicator in sensory characteristics was more sensitive to the freshness of *T. matsutake*.

The changes of physical and chemical characteristics can be divided into three periods, S_1_, S_2_, and S_3_, respectively, to interpret the physiological changes of *T. matsutake*. The changes of pH dramatically increased in the beginning during period S_1_, lasting about two days. This was because the cap of *T. matsutake* would spread after harvest, like many mushrooms. The changes of SSC and MC decreased in the period S_1_, lasting about four days. In period S_2_, the changes of pH, SSC, and MC were stable. The pH and MC all slowly decreased, and the SSC maintained stable over time. In the period S_3_, the fluctuation of pH suggested a great deal more volatility, which also appeared in the changes of SSC and MC. With the preservation time flowing, individual quality differences of *T. matsutake* greatly increased.

The three sensory characteristics—color, odor, and stipe hardness—showed great significance with the remaining shelf life, thus becoming the variables of prediction models in this paper by the correlation analysis. By analyzing the BP neural network prediction model results, the prediction data was over the error range in the fourth day. This was because there were only four levels of sensory evaluation, which led to the same scores on several consecutive days in our experiments. However, there was the possibility of dramatic changes on postharvest characteristics and quality of *T. matsutake* from the second day to the fourth day because of its cap spread process, which showed in our analysis on the changes of physical and chemical characteristics. Future research should focus on the more specific changes of physical and chemical characteristics and even microbial characteristics of *T. matsutake* after harvest, and the time interval for measuring these indicators should be shortened to once or twice a day.

## Figures and Tables

**Figure 1 foods-08-00136-f001:**
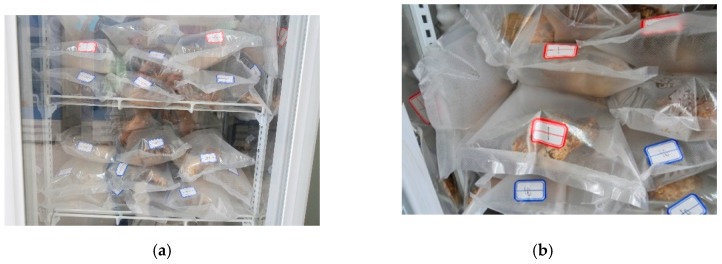
Different modified atmosphere packaging conditions of *T. matsutake* in the experiment. (**a**) The test packing environment. (**b**) Packing details of this test in Group.2 and Group.3.

**Figure 2 foods-08-00136-f002:**
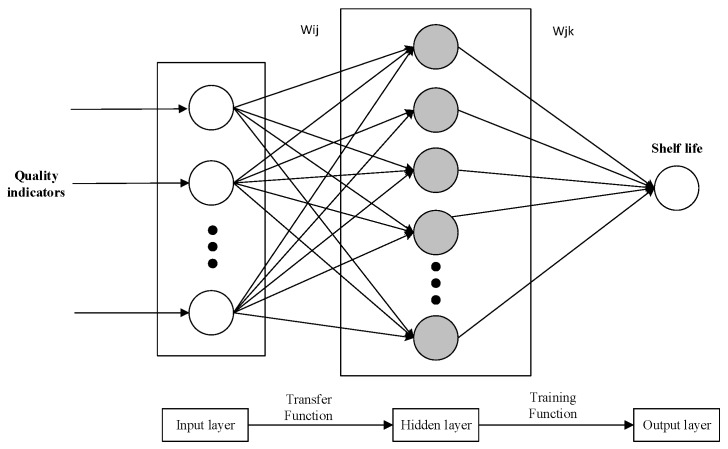
Topology of Back Propagation (BP) neural network.

**Figure 3 foods-08-00136-f003:**
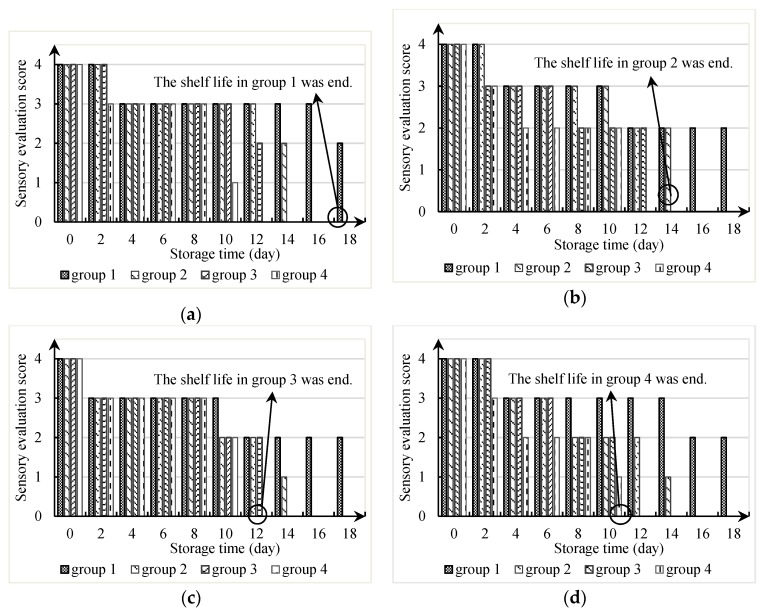
Sensory evaluation score changes of fresh Tibetan *T. matsutake*. (**a**) The cap hardness evaluation. (**b**) The stipe hardness evaluation. (**c**) The color evaluation. (**d**) The odor evaluation.

**Figure 4 foods-08-00136-f004:**
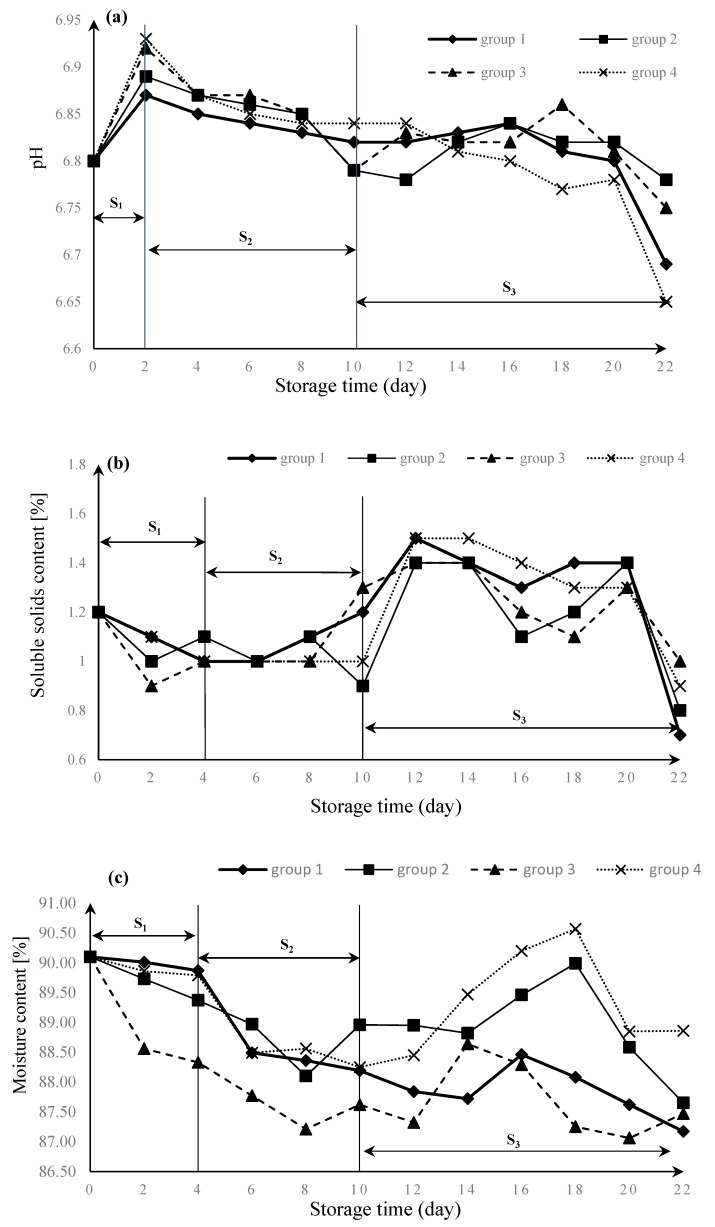
Changes in (**a**) pH, (**b**) percentage of soluble solids content (SSC), and (**c**) moisture content (MC) of fresh Tibetan *T. matsutake*.

**Figure 5 foods-08-00136-f005:**
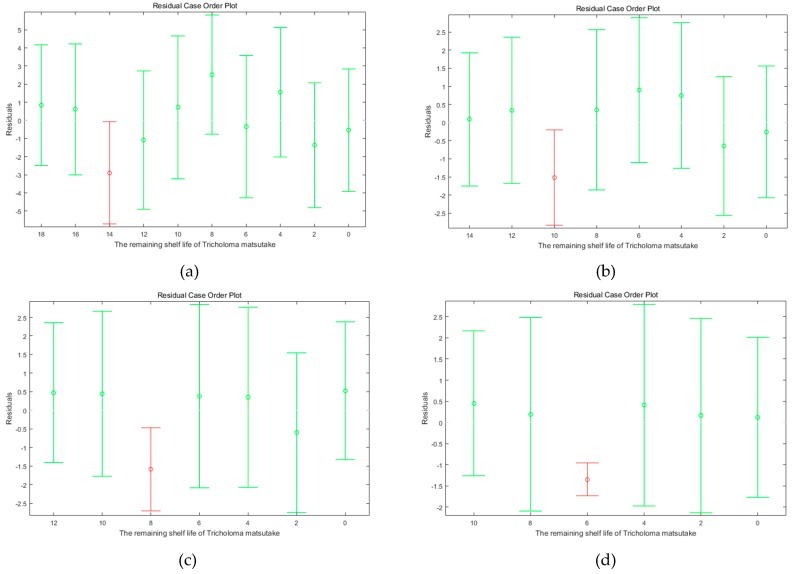
The residual case order plot of BP neural network prediction results under the MAP conditions: (**a**) $1\%O_{2}+21\%CO_{2}+78\%N_{2}$; (**b**) $5\%O_{2}+17\%CO_{2}+78\%N_{2}$; (**c**) $10\%O_{2}+12\%CO_{2}+78\%N_{2}$; (**d**) Air group.

**Table 1 foods-08-00136-t001:** Gas ratio of test environment.

Group	O_2_	CO_2_	N_2_	Air
1	1%	21%	78%	-
2	5%	17%	78%	-
3	10%	12%	78%	-
4	-	-	-	100%

**Table 2 foods-08-00136-t002:** *T. matsutake* sensory quality scoring standard.

Score	Hardness	Color	Odor
4	The cap is elastic, and the stipe is hard.	The cap is fresh, and its color is shining chestnut-colored. The stipe is ivory, no browning.	It has strong fragrance.
3	The cap is less elastic, and the stipe is less hard.	Normal color, slight browning.	Normal, no peculiar smell.
2	The cap and stipe begin to soften.	Moderate browning.	Slightly peculiar smell.
1	The cap and stipe begin to severely soften	Severe browning, mildew appearing.	Serious peculiar smell.

**Table 3 foods-08-00136-t003:** Correlation between quality and physicochemical indexes of *T. matsutake*.

Groups	Cap Hardness	Stipe Hardness	Color	Odor	pH	Soluble Solids Content	Moisture Content
1	0.81 *	0.93 **	0.90 **	0.92 **	0.26	−0.71	0.84 *
2	0.86*	0.96 **	0.91 **	0.96 **	0.47	−0.46	0.71
3	0.89*	0.92 **	0.89 *	0.94 **	0.33	−0.58	0.88 *
4	0.82*	0.96 **	0.92 **	0.93 **	0.11	0.83 *	0.94 **

* indicates that there is the linear relationship, ** indicates that the linear correlativity is prominent, and the correlation is not obvious if it is not marked.

**Table 4 foods-08-00136-t004:** The remaining shelf life of BP neural network prediction models.

Title	Group 1	Group 2	Group 3	Group 4
Actual Data	Prediction Data	Absolut Error	Actual Data	Prediction Data	Absolut Error	Actual Data	Prediction Data	Absolute Error	Actual Data	Prediction Data	Absolute Error
0	18.00	18.03	0.03	14.00	13.49	0.51	12.00	12.00	0.00	10.00	9.94	0.06
2	16.00	15.99	0.01	12.00	11.87	0.13	10.00	10.00	0.00	8.00	7.93	0.07
4	14.00	10.68	3.32	10.00	8.14	1.86	8.00	6.00	2.00	6.00	4.64	1.36
6	12.00	10.68	1.32	8.00	8.14	0.14	6.00	6.00	0.00	4.00	4.64	0.64
8	10.00	10.68	0.68	6.00	6.81	0.81	4.00	4.00	0.00	2.00	2.64	0.64
10	8.00	10.68	2.68	4.00	4.79	0.79	2.00	1.08	0.92	0.00	0.85	0.85
12	6.00	6.00	0.00	2.00	1.52	0.48	0.00	0.24	0.24			
14	4.00	5.09	1.09	0.00	0.04	0.04						
16	2.00	1.36	0.65									
18	0.00	0.38	0.38

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
