# Peer review of "Quality Characteristics Analysis and Remaining Shelf Life Prediction of Fresh Tibetan Tricholoma matsutake under Modified Atmosphere Packaging in Cold Chain"

_foods, 2019, doi:10.3390/foods8040136_

Round 1
Reviewer 1 Report
The manuscript present a research on quality characteristics and shelf-life prediction of Tricholoma matsutake mushrooms. Although interesting as idea, the manuscript reports poorly described methods and also the results and discussion section needs to be improved.
Below you will find specific comments.
L48, L53: its instead of their
L64 organism instead of organisms. Continues instead of continue.
L113-114: the authors should specify how cleaning, grading and sterilizing treatments were performed.
L128: the technological characteristics of the vacuum fresh keeping bag used to pack the Tricholoma matsutake should be specified.
Paragraph 2.2. How were the different gas mixture realized? How were the bags filled and sealed? Which kind of machine was employed? Please specify.
Paragraph 2.3. How were the measurements performed?
How many panelists for the sensory evaluation? Did they receive a form to be filled?
L150 what is PVP? Please specify.
Was the colour experimentally measured?
How was the correlation analysis performed?
Which was the meaning of the determination of total soluble solids? A part from the colorymetric analysis, how was the determination performed? How many grams of samples were analyzed?
No determination of microbiological parameters?
L225: whose instead of whoes; scores instead of score
L227: fragrance instead of fragranc
L245: figure legend. pH instead of Ph. Maybe Tibetan instead of Tebiten?
L249: ramatic?
L252: than instead of then
How could the pH dynamics (first increase, then decrease during time) be explained?
How do the authors explain the different trend observed for groups 2 and 4 (increasing) and groups1 and 3 (decreasing) MC in days 14-18?
It is not so surprising that total soluble content increases as moisture content decreases!
L315: first instead of fisrt.
L334 indicators instead of indicaters
As the authors already individuated differences in the evolution of quality parameters (S1, S2 and S3 periods), they should better explain the advantages of using BP-neural network to predict the remaining shelf-life. In 3.3.4 is reported that the forth day is crucial, however the analysis does not seem to predict the remaining shelf-life.
The conclusion section should not be a resume of the results.
Author Response
Response to Reviewer 1 Comments
Dear Editor,
Thank you very much for considering our paper (Manuscript Number: foods-480270) “Quality characteristics analysis and remaining shelf life prediction of fresh Tibetan Tricholoma matsutake under Modified Atmosphere Packaging in cold chain”. We are also very thankful to the constructive comments from the reviewers. We have revised the manuscript according to the reviewers’ comments. Please see below our responses to the reviewer comments and will have also indicated our revisions in the manuscript.
Point 1: L48, L53: its instead of their
L64: organism instead of organisms. Continues instead of continue.
L225: whose instead of whoes; scores instead of score
L227: fragrance instead of fragranc
L245: figure legend. pH instead of Ph. Maybe Tibetan instead of Tebiten?
L249: ramatic?
L252: than instead of then
L315: first instead of fisrt.
L334: indicators instead of indicaters
Response 1: We thank the reviewer for the comments to our manuscript. These spelling errors have been corrected in the article.
Point 2: L113-114: the authors should specify how cleaning, grading and sterilizing treatments were performed.
Response 2: We thank the reviewer for the comments to our manuscript. We have revised the cleaning, grading and sterilizing treatments to pre-cooling, sorting and cleaning treatments. The process was described detailedly in line 111 to line 120.
Point 3: L128: the technological characteristics of the vacuum fresh keeping bag used to pack the Tricholoma matsutake should be specified.
Response 3: We thank the reviewer for the comments to our manuscript. We have added the technological characteristics of the vacuum fresh keeping bag in line 122.
Point 4: Paragraph 2.2. How were the different gas mixture realized?
How were the bags filled and sealed? Which kind of machine was employed? Please specify.
Response 4: We thank the reviewer for the comments to our manuscript. The standard gases of multi-component were brought from a standard gas factory in local (Linzhi, Tibet).
The bags filled and sealed process was specified in line 131 to line 134, which also included the type of machines.
Point 5: Paragraph 2.3. How were the measurements performed?
How many panellists for the sensory evaluation? Did they receive a form to be filled?
Response 5: We thank the reviewer for the comments to our manuscript. We have added the measurements in line 148 to line 149. There are 5 panellists for the sensory evaluation, which included the local gatherer, acquirer, cook and two consumers. They need to fill a form separately each time.
Point 6: L150 what is PVP? Please specify.
Response 6: We thank the reviewer for the comments to our manuscript. We have added the explanations about PVP in line156.
Point 7: Was the colour experimentally measured?
A part from the colorimetric analysis how was the determination performed?
Response 7: We thank the reviewer for the comments to our manuscript. The colour in the article was part of sensory evaluation. In the experiment, we did measure the chromatic aberration (L*, a*, b*) by a spectrophotometer (Konica Minolta Holdings Inc., Tokyo, Japan). But data differences were too obvious to analyze. Therefore, we give up the chromatic aberration data. In previous studies, some researchers showed that the chromatic aberration of Tricholoma matsutake would change with its shelf life decreasing through experiments. However, we couldn’t analyse a result from our data. There may be problems with both measuring instruments and methods. We will continue to check and improve in the next experiment if there is an opportunity.
Point 8: How was the correlation analysis performed?
Response 8: We thank the reviewer for the comments to our manuscript. We have added the correlation analysis performed process in line 164 to line 166.
Point 9: Which was the meaning of the determination of total soluble solids?
Response 9: We thank the reviewer for the comments to our manuscript. In this paper, we are sorry for that we didn't unify the total soluble solids (TSS) and soluble solids content (SSC). We have unified it as soluble solids content (SSC) in our paper. In our introduction section, we have stated that some researchers considered SSC was a main shelf life influence factors of Tricholoma matsutake. Therefore, we measured SSC and analyzed the relationship between SSC and its remaining shelf life.
Point 10: How many grams of samples were analyzed?
Response 10: We thank the reviewer for the comments to our manuscript. The grams of samples have been introduced in Paragraph 2.3., which included every indicator measurement method in this paper.
Point 11: No determination of microbiological parameters?
Response 11: We thank the reviewer for the comments to our manuscript. Yes. We don’t measure the microbiological parameters.
Point 12: How could the pH dynamics (first increase, then decrease during time) be explained?
Response 12: We thank the reviewer for the comments to our manuscript. We have explained this change in line 254 to line 256. The cap of Tricholoma matsutake would go through a spread period after harvest like many kinds of mushrooms because of respiration. Before the cap spread, the pH dynamics increased, and the pH dynamics decreased after the cap spread.
Point 13: How do the authors explain the different trend observed for groups 2 and 4 (increasing) and groups1 and 3 (decreasing) MC in days 14-18?
Response 13: We thank the reviewer for the comments to our manuscript. We have explained these changes in line 275 to line 281. On the one hand, individual differences did exist between Tricholoma matsutakes in the experiments. On the another hand, In days 14-18, we considered that groups 2,3,4 had stopped its shelf life, and the pH, SSC and MC changes in these days were considered as its deterioration period, which was not the main contents in this paper. Like in group 4, the shelf life had stopped on the tenth day through the sensory evaluation. After its shelf life, Tricholoma matsutakes would appear autolysis phenomenon, which could make the MC increase. Besides, there are few researches about Tricholoma matsutakes, and these researchers didn’t conclude that which indicator could be as the signal to judge the end of its shelf life. If consumer don’t want to buy it, it will loses its commercial value and its shelf life stops. So in this paper, we used the sensory evaluation (or called consumer evaluation) to be as the end standard of its shelf life.
Point 14: As the authors already individuated differences in the evolution of quality parameters (S1, S2 and S3 periods), they should better explain the advantages of using BP-neural network to predict the remaining shelf-life.
Response 14: We thank the reviewer for the comments to our manuscript. We have explained the advantages of using BP-neural network in line 95 to line 99 and line 353 to line 356.
Point 15: In 3.3.4 is reported that the forth day is crucial; however the analysis does not seem to predict the remaining shelf-life.
Response 15: We thank the reviewer for the comments to our manuscript. We predicted the remaining shelf life by BP neural network and the results were showed in the Table 4. We report that the forth day is crucial through analysing the predicting results that the forth day’s predicting result was abnormal.
Point 16: The conclusion section should not be a resume of the results.
Response 16: We thank the reviewer for the comments to our manuscript. We have revised our conclusion in the section 4 from line 348 to line 382.

Reviewer 2 Report
matsutake MAP/Cold chain in Tibet
Requires a careful review for typos and misspellings. Tibetan was misspelled in 5 different ways.
Abstract should mention the temperature parameters for the study (4 C) and RH% (90)
line 13: is difficult to save.
"save" is not a postharvest term. I suggest instead "is difficult to store" or "has an extremely short shelf life".
line 50: which makes it easy to infection
please revise to read: which makes it highly susceptible to infection
line 124-125: and the humidity was the same content as air humidity.
Please provide the RH% for the study.
What is the RH% of the air? need to include it here.
(line 211: and 90% RH in polyethylene pouches) ???
line 127: table 1 -- Air also contains these gases.
what is the % O2, CO2 and N2 in the air (in Tibet)? was it measured?
line 145: the final sensory score was determined to be more than 50% agreed by the evaluation team.
Why did you not use the average of the 5 scores? It seems like you are discarding useful information.
line 213: Tebitan
Tibetan typo
line 222: these four kinds indicaters
please revise to read: these four kinds of indicators
line 224: Tetiten
Tibetan typo
line 225: whoes
typo: correct to whose
line 227: fragranc
typo: correct to fragrance
line 245: Tebiten, another Tibetan typo.
Ph (should be pH)
line 249: ramatic
??? dramatically ???
line 338 dramatic
dramatically
Line 323- end: Conclusions require improvements:
Conclusions are focused on the indicators of quality (stipe hardness, color and odor), but there is no mention of the underlying causes of quality changes-- the oxygen concentration effects. The conclusions should at least describe the best MAP conditions for matsutake, based on the results of the study. Temperature management, which is "in cold chain" in the title, is barely mentioned at all.
In addition, it is very likely that larger differences are due to temperature and RH% effects than to MAP. Future studies should measure and report storage temperature along with each day's quality assessments. (All refrigeration systems fluctuate a few degrees from day to day, and can be affected by hot or cold weather, power outages, etc). The same study conducted for matsutake stored in the best MAP conditions (low oxygen) but under two temperature regimes (4C and 1C) and two RH% regimes (90% and 95%) may provide much more information and lead to better conclusions on improving shelf life by using MAP and the cold chain.
Author Response
Response to Reviewer 2 Comments
Dear Editor,
Thank you very much for considering our paper (Manuscript Number: foods-480270) “Quality characteristics analysis and remaining shelf life prediction of fresh Tibetan Tricholoma matsutake under Modified Atmosphere Packaging in cold chain”. We are also very thankful to the constructive comments from the reviewers. We have revised the manuscript according to the reviewers’ comments. Please see below our responses to the reviewer comments and will have also indicated our revisions in the manuscript.
Point 1: Requires a careful review for typos and misspellings. Tibetan was misspelled in 5 different ways. Line 213: Tebitan, Tibetan typo; line 224: Tetiten, Tibetan typo; line 245: Tebiten, another Tibetan typo.
Response 1: We thank the reviewer for the comments to our manuscript. These spelling errors have been corrected in the article.
Point 2: Abstract should mention the temperature parameters for the study (4 C) and RH% (90)
Response 2: We thank the reviewer for the comments to our manuscript. We have added them in the abstract.
Point 3: Line 13: is difficult to save. "save" is not a postharvest term. I suggest instead "is difficult to store" or "has an extremely short shelf life".
Line 50: which makes it easy to infection.
Please revise to read: which makes it highly susceptible to infection.
Line 222: these four kinds indicaters.
Please revise to read: these four kinds of indicators.
Response 3: We thank the reviewer for the comments to our manuscript. We have revised these problem in our article based on your suggestion.
Point 4: Line 124-125: and the humidity was the same content as air humidity.
Please provide the RH% for the study.
What is the RH% of the air? need to include it here.
(line 211: and 90% RH in polyethylene pouches) ???
Response 4: We thank the reviewer for the comments to our manuscript. In our experiments, the RH was set at about 90% to preserve the fresh Tibetan Tricholoma matsutake in the vacuum fresh keeping bags made of polyethylene (PV). We increased the humidity in the bags measured by a needle RH sensor (HUATO Shenzhen China).
Point 5: Line 127: table 1 -- Air also contains these gases.
What is the % O2, CO2 and N2 in the air (in Tibet)? Was it measured?
Response 5: We thank the reviewer for the comments to our manuscript. We didn’t measure the % of O2, CO2 and N2 in the air (in Tibet). Air group was set as a controlled trial. In group 1, 2 and 3, the content of O2 were lower than it is in air.
Point 6: Line 145: the final sensory score was determined to be more than 50% agreed by the evaluation team. Why did you not use the average of the 5 scores? It seems like you are discarding useful information.
Response 6: We thank the reviewer for the comments to our manuscript. We made a statement mistake. To be exactly, we chose the median of evaluation scores as the sensory evaluation result. We have revised it in line 151. In this paper, we used the sensory evaluation (or called consumer evaluation) to be as the end standard of its shelf life. Because there are few researches about Tricholoma matsutakes, and these researchers didn’t conclude that which indicator could be as the signal to judge the end of its shelf life.
Point 7: Line 225: whoes, typo: correct to whose
Line 227: fragranc, typo: correct to fragrance
Ph (should be pH)
Line 249: ramatic, ??? dramatically ???
Line 338 dramatic, dramatically
Response 7: We thank the reviewer for the comments to our manuscript. These spelling errors have been corrected in the article.
Point 8: Line 323- end: Conclusions require improvements:
Conclusions are focused on the indicators of quality (stipe hardness, color and odor), but there is no mention of the underlying causes of quality changes-- the oxygen concentration effects. The conclusions should at least describe the best MAP conditions for matsutake, based on the results of the study. Temperature management, which is "in cold chain" in the title, is barely mentioned at all.
Response 8: We thank the reviewer for the comments to our manuscript. We have revised our conclusion in the section 4. from line 357 to line 359.
In addition, our experiments were constantly carried out in the cold chain environment, which included our precooling process, the process of transporting to the laboratory, and the process of storage in thermostat at low temperature.
Point 9: In addition, it is very likely that larger differences are due to temperature and RH% effects than to MAP. Future studies should measure and report storage temperature along with each day's quality assessments. (All refrigeration systems fluctuate a few degrees from day to day, and can be affected by hot or cold weather, power outages, etc). The same study conducted for matsutake stored in the best MAP conditions (low oxygen) but under two temperature regimes (4C and 1C) and two RH% regimes (90% and 95%) may provide much more information and lead to better conclusions on improving shelf life by using MAP and the cold chain.
Response 9: We thank the reviewer for the comments to our manuscript. Our experiments have measured the quality characteristics of fresh Tibetan Tricholoma matsutake in different temperatures including 0℃, 4℃, 10℃ and 20℃. We wrote another article about these results and the results showed that the best preservation environment was at 4℃. So the experiments in our article were carried out in 4℃. The RH% regimes was about 90% according to the previous researches of Wang Yue & Xue Wei (2012) which studied the effect of different temperature and humidity on preservation of Tricholoma matsutake in Chinese. We set different kinds experiments about different temperatures, freshness-retention chemicals, packaging and storage conditions and some other experiments about nutritional components of fresh Tibetan Tricholoma matsutake. We wrote these experiments’ results in other articles. In some experimental environments, the shelf life of Tricholoma matsutake was shorter, like in the temperature of 20 ℃, which we measured the quality indicators each day. And in the temperature of 4 ℃, its shelf life is longer, we measured the quality indicators once every other day. However, there were a little bit less data in the group 3 and 4 in the paper.

Reviewer 3 Report
The article undoubtedly raises an issue that is important for the economy of the Province of Yunnan. This makes the underdevelopment of the experiment all the more relevant. It is known that cap mushroom fruiting bodies are extremely sensitive to touch, which damages the structure of the hat and leads to the leakage of cellular content.
The pictures in the manuscript show that the mushrooms are stored in loose bags, which are stored one on top of the other, which can lead to crushing of the fruiting body of the mushroom.
The mushrooms should be examined while being packed in rigid molds, which should then be placed in bags and hermetically sealed.
Analytical methods are imprecisely described, and additionally, the color measurement should be carried out in an objective and instrumental way with the use of a color meter.
The use of a neural network will not improve the quality of the article if the basic data has been collected in a questionable way.
I recommend reworking the material and, after obtaining reliable data, redoing the analysis. Such an article will then be an article of high cognitive and economic value.
Author Response
Response to Reviewer 3 Comments
Dear Editor,
Thank you very much for considering our paper (Manuscript Number: foods-480270) “Quality characteristics analysis and remaining shelf life prediction of fresh Tibetan Tricholoma matsutake under Modified Atmosphere Packaging in cold chain”. We are also very thankful to the constructive comments from the reviewers. We have revised the manuscript according to the reviewers’ comments. Please see below our responses to the reviewer comments and will have also indicated our revisions in the manuscript.
Point 1: The article undoubtedly raises an issue that is important for the economy of the Province of Yunnan. This makes the underdevelopment of the experiment all the more relevant. It is known that cap mushroom fruiting bodies are extremely sensitive to touch, which damages the structure of the hat and leads to the leakage of cellular content.
The pictures in the manuscript show that the mushrooms are stored in loose bags, which are stored one on top of the other, which can lead to crushing of the fruiting body of the mushroom.
The mushrooms should be examined while being packed in rigid molds, which should then be placed in bags and hermetically sealed.
The use of a neural network will not improve the quality of the article if the basic data has been collected in a questionable way. I recommend reworking the material and, after obtaining reliable data, redoing the analysis. Such an article will then be an article of high cognitive and economic value.
Response 1: We thank the reviewer for the comments to our manuscript. Tricholoma matsutake industry is very important to forestry economy in Tibet and the quality of Tricholoma matsutake in Tibet is the best in China. Our experiments in this article carried out in Lingzhi City, Tibet Autonomous Region, was to study the quality characteristics of fresh Tricholoma matsutake.
We are sorry that the pictures in the manuscript showing the mushrooms caused you the misunderstanding that Tricholoma matsutakes were stored in loose bags. The fresh-keeping bags containing the fresh Tricholoma matsutakes were filled with multi-component calibration gases. When they were put in the thermostats, the bags would squeeze each other, but the Tricholoma matsutake in the bags will not be squeezed, because there was enough space for the Tricholoma matsutakes. If we put the rigid molds to contain the bags, this method will block the spread of temperature in our experimental thermostats.
In our experiments, operating environment is controlled in accordance with standard conditions. Instruments are operated in accordance with operational specifications. Operators have been trained before the experiment, who have also done other similar experiments. We set different kinds experiments about different temperatures, freshness-retention chemicals, packaging and storage conditions and some other experiments about nutritional components of fresh Tibetan Tricholoma matsutake. We wrote these experiments’ results in other articles. Some of them have been accepted by other journals.
Point 2: Analytical methods are imprecisely described, and additionally, the color measurement should be carried out in an objective and instrumental way with the use of a color meter.
Response 2: We thank the reviewer for the comments to our manuscript. The color in the article was part of sensory evaluation. In the experiment, we did measure the chromatic aberration (L*, a*, b*) by a spectrophotometer (Konica Minolta Holdings Inc., Tokyo, Japan). However, data differences were too obvious to analyze. Therefore, we give up the chromatic aberration data. In previous studies, some researchers showed that the chromatic aberration of Tricholoma matsutake would change with its shelf life decreasing through experiments. However, we couldn’t analyse a result from our data. We will continue to check and improve in the next experiment if there is an opportunity.

Round 2
Reviewer 1 Report
The authors revised the manuscript, adding data and explanations required to make it more complete. Nevertheless, the english language still need to be revised and correct in several points. In my opinion this step of text editing and language revision is still necessary.
Reviewer 3 Report
The Authors have improved their manuscript. I don not have further comments.